# Effectiveness of High-Intensity Laser Therapy Plus Ultrasound-Guided Peritendinous Hyaluronic Acid Compared to Therapeutic Exercise for Patients with Lateral Elbow Tendinopathy

**DOI:** 10.3390/jcm11195492

**Published:** 2022-09-20

**Authors:** Raffaello Pellegrino, Teresa Paolucci, Fabrizio Brindisino, Paolo Mondardini, Angelo Di Iorio, Antimo Moretti, Giovanni Iolascon

**Affiliations:** 1Department of Scientific Research, Campus Ludes, Off-Campus Semmelweis University, 6912 Lugano, Switzerland; 2Physical Medicine and Rehabilitation, Department of Oral Medical Science and Biotechnology, “G d’Annunzio” University Chieti-Pescara, 66100 Chieti, Italy; 3Department of Medicine and Health Science “Vincenzo Tiberio”, University of Molise c/o Cardarelli Hospital, C/da Tappino, 86100 Campobasso, Italy; 4Department of Sport Science, 40100 Bologna, Italy; 5Department of Innovative Technologies in Medicine & Dentistry, “G. d’Annunzio” University Chieti-Pescara, 66100 Chieti, Italy; 6Department of Medical and Surgical Specialties and Dentistry, University of Campania “Luigi Vanvitelli”, 80138 Naples, Italy

**Keywords:** hyaluronic acid, high-intensity laser therapy, lateral elbow tendinopathy, Patient-Rated Tennis Elbow Evaluation, muscle strength, disability, ultrasound, rehabilitation

## Abstract

Lateral elbow tendinopathy (LET) is a common painful musculoskeletal disorder. Several treatments have been proposed to provide pain reduction and functional recovery, including laser therapy, hyaluronic acid peritendinous injection (Hy-A), and therapeutic exercise (TE). Our study aims to assess the effectiveness of a combined approach with high-intensity laser therapy (HILT) and Hy-A injections compared to TE on pain, muscle strength, and disability in patients with painful LET. A retrospective longitudinal study was carried out by consulting the medical records of patients with a diagnosis of painful LET formulated by clinical and instrumental findings that received functional evaluations, including the Patient-Rated Tennis Elbow Evaluation (PRTEE) and muscle strength measurement at least four times: T0 (“baseline”), 1-month (T1), 3-month (T2), and 6-month follow-ups (T3). Medical records of 80 patients were analyzed. In the HILT + HyA group, the Peak-strength (*p* < 0.001) and mean strength (*p* < 0.001) significantly increased compared to the TE group between study times. For the PRTEE-total-score as for the subscales, the HILT + HyA group reported statistically significant reductions only for the comparisons of baseline versus T1 and baseline versus T2. No serious adverse events occurred. Our findings suggest that Hy-A associated with HILT might be more effective than TE for people with LET in the short–medium term.

## 1. Introduction

“Tennis elbow”, or lateral elbow tendinopathy (LET), is a noninflammatory condition that affects the tendon insertion or myotendinous junction of wrist muscle extensors [1], causing subacute and chronic symptoms of pain at the lateral epicondyle and disability of the elbow and sometimes of the entire upper limb. LET occurs in between 1% and 3% of the population and typically affects subjects between 30 and 60 years without gender difference [1]. Several determinants were reported to be associated with LET, such as working procedures characterized by the long-term repetitive forearm and hand movement [2], the excessive neuronal activity by nociceptors that derive from the radial nerve leading to axonal sprouting of the free nerve endings and peripheral sensitization, smoking habits, and metabolic factors, such as estrogen decline, hypercholesterolemia, and obesity [1]. LET is a degenerative overuse process of the extensor carpi radialis brevis and of the common extensor tendon [3] characterized by histological micro-rupture, vascular proliferation, and hyaline degeneration without inflammatory cells infiltrating within the tendon tissue [4]. The main clinical manifestation is hyperalgesia during active range of motion of the elbow and at the palpation in the lateral epicondyle area, which is exacerbated by prono-supination of the forearm [5]. Moreover, LET patients complain of painful handgrip with consequent functional limitation, disability in activities of daily living, time lost at work, and poor quality of life [6]. LET is usually considered a self-limiting condition, with the majority of patients recovering in 6–24 months [7], even if some clinical reports noticed symptom recurrence persisting for many years [8]. Several conservative approaches have been proposed to manage LET, including pharmacological therapy, systemic and/or local treatments (corticosteroid injections, botulinum toxin, hyaluronic acid, autologous blood, and platelet-rich plasma) [9], therapeutic exercise (TE), physical modalities, elbow braces, acupuncture, and watchful waiting [8]. Surgery is usually recommended for those patients with persistent pain and disability after a course of conservative therapy [10].

However, no consensus about the best treatment for improving pain and function in people with LET has been reached. Among rehabilitative approaches, manual therapy and TE showed positive effects on people with LET in terms of pain relief and increased tendon strength [11,12]. Among physical modalities commonly used for LET, laser therapy was previously demonstrated to improve grip strength, pain, and functional ability at midterm follow-up (5 to 26 weeks) compared to placebo [8]. The studies on laser treatment in LET have focused mainly on the efficacy of low-level laser therapy (LLLT), and few studies have investigated the efficacy of high-intensity laser therapy (HILT) [8]. However, a meta-analysis highlighted how HILT could be more effective than LLLT in terms of pain control, stiffness, and function in degenerative musculoskeletal conditions [13].

Injection therapy is widely used for the treatment of patients with LET [14]. In particular, peritendinous hyaluronic acid (Hy-A) injection seems to be an effective therapeutic option for pain control and functional improvement in these patients.

We hypothesized that a combined approach with HILT plus peritendinous Hy-A injection could improve pain control and functional recovery, considering the analgesic effect and stimulation of collagen synthesis attributable to HILT [15] and the Hy-A-related enhancement of the activity of the fibroblasts, including their adhesivity, extracellular matrix synthesis, and proliferation [16,17]. 

Therefore, the main objective of the study is to assess the effectiveness of the combination of HILT and Hy-A peritendinous injection on pain relief, improvement of muscle strength, functional ability, and quality of life, in a mid-long-term period, compared to TE in patients with painful LET.

## 2. Materials and Methods

### 2.1. Study Design

A retrospective cohort clinical study was conducted by the Declaration of Helsinki and the STROBE guidelines [18]. The study protocol was planned at the Department of Clinical Research, Ludes Campus, Luganoff Campus of Semmelweis University of Budapest, and written informed consent was obtained from all participants for treatment and data processing. From June 2021 to June 2022, medical records of patients with LET that were treated at the Chiparo Physical Medicine and Rehabilitation Clinic in Lecce, Italy, were consulted and selected by a specialized nurse blinded for the main outcomes of the study.

### 2.2. Population

We included medical records of patients between 30 and 65 years with elbow pain for at least 2 weeks and the following clinical and instrumental features: (1) no other sources of elbow pain (e.g., cervical radiculopathy); (2) minimal pain at rest (Patient-Rated Tennis Elbow Evaluation, PRTEE, pain subscale < 3); (3) positive wrist extension tests against resistance (Cozen test) [19]; (4) positive palpation test of the epicondyle; (5) ultrasound (US) evaluation showing thickening and heterogeneous echo structure of the common extensor tendon, as well as increased blood flow under Doppler. We did not consider clinical records of patients receiving regular painkillers, previous elbow injections, specific rehabilitation treatments in the previous two months, and those with US-scan-confirmed injury or rupture of any extensor tendon of the carpus or with previous elbow surgery. Additional exclusion criteria were the presence of contraindications to laser therapy and Hy-A injection, such as drug allergy, epilepsy, coagulopathies or anticoagulant therapy, neoplasms, or pregnancy. Patients with fibromyalgia, enthesitis due to seronegative arthritis, and rheumatoid arthritis were also excluded. Finally, patients who had skin infections at the injection site, systemic symptoms (e.g., fever), or were carriers of artificial cardiac pacemakers were also excluded.

### 2.3. Outcomes

We included medical records of patients who filled the PRTEE questionnaire [20] and who received the measurement of maximum (peak) and mean grip strength by an electronic dynamometer (Activeforce2 Sixtus Prato (PO) Italia) during contraction against the resistance of wrist extensors with the elbow flexed at 90° (Cozen test). Handgrip strength protocol provided 3 maximal contraction trials lasting 5 s each, interspersed with 15 s of rest. The PRTEE is a 15-item questionnaire assessing pain (5 items) and the degree of difficulty in performing various activities (6 specific and 4 usual activities) due to LET over the preceding week [20]. The same functional assessments were performed at different follow-ups: T1 (30 days), T2 (90 days), and T3 (180 days) after the end of treatment.

### 2.4. Interventions

#### 2.4.1. HILT Plus Hyaluronic Acid (HILT + Hy-A) Group

Our HILT protocol consisted of 10 daily sessions using a LASERIX PRO device (GN med), administered via a fixed tip with a 30 mm spacer. Each HILT session had a total duration of 13 min and was divided into 3 phases. In the first phase lasting 7 min, a frequency of 18 Khz, peak power of 600 W, and 226 Joules of energy was delivered. In the second phase lasting 3 min, a frequency of 14 Khz, peak power of 900 W, and 226 Joules of energy was delivered. In the third phase lasting 3 min, a frequency of 10 Khz, peak power of 1200 W, and 226 Joules of energy was delivered. At the end of the 10 HILT sessions, and on day 7 and day 14 after the last therapeutic session, patients received US-guided (SonoSite M-Turbo ultrasound system with a 6–15 Mhz linear probe) Hy-A injections in the peritendinous area of the elbow epicondyle by the same physiatrist expert in US technique and US-guided injection (Appendix A). The injections were conducted using a pre-filled syringe of 20 mg in 2 mL of linear Hy-A sodium salt with a molecular weight of 500–730 kDa. The needles used were 25G 25 mm. No anesthetic drugs were administered after the injections. During the therapeutic procedure with laser and Hy-A injections, patients were seated, and the affected elbow was positioned at 90° of flexion and in slight pronation. The skin was disinfected with a chlorhexidine wipe. All subjects enrolled in this group were not prescribed any therapeutic exercise and did not receive any other physical or pharmacological therapy.

#### 2.4.2. Therapeutic Exercise Group

Our TE protocol (eccentric exercises series and static stretching exercise) consisted of three/per week clinical supervised sessions for 4 weeks for a total of 12 sessions. Eccentric exercises for LET were performed with the elbow supported on the bed in full extension, forearm in pronation, wrist in maximal extension, and hand hanging over the edge of the bed. In this position, patients were told to flex their wrist slowly until full flexion was achieved and then return to the starting position. Patients were instructed to continue with the exercise even if they experienced mild pain. However, they were instructed to stop the exercise if disabling pain occurred. Each session consisted of three sets of 10 repetitions with at least a 1 min rest interval between each set. When patients were able to perform the eccentric exercises without experiencing any pain or discomfort, the load was increased using free weights or therabands. The starting and final positions of eccentric exercises, the increase in the load, and the degree of mild or disabling pain could not be standardized but were tailored to each patient [11]. The static stretching exercises for LET were performed slowly with the elbow in extension, forearm in pronation, wrist in flexion, and ulnar deviation according to the patient’s tolerance to achieve the best stretching position result for the ECRB tendon. This position was held for 30–45 s before and after each set of eccentric exercises [11]. 

#### 2.4.3. Treatments-Related Adverse Events

We reported adverse events (AEs) in both groups by consulting the clinical records of included patients [21].

### 2.5. Sample Size

The sample size was calculated for the comparison of the two study groups at a 6-month follow-up for the PRTEE—total score. We assumed as significant minimal change a difference of at least five points in the PRTEE at each follow-up. We also settled for standard deviations of 10. A sample size of N = 36 per study arm (72 overall) provides more than 90% power (alpha = 0.05, two-tailed) applying linear mixed models, with an intraclass-correlation between measures ρ = 0.50. To account for 10% attrition during the study period (“dropouts”), we planned to recruit 40 medical records per study group (80 overall) at baseline. Similar values were obtained in the assessment of the sample size according to peak muscle strength [22].

### 2.6. Statistical Analysis

Data were reported as mean ± standard error (S.E.) for continuous variables and as absolute number and percentage for dichotomous variables; differences between groups were assessed with analysis of variance and chi-square test, respectively. To assess the variation of the PRTEE subscale as the total score, peak, and mean muscle strength, linear mixed models (LMMs) were applied [23]. Intercept and time had a random component. The advantage of this approach is that it increases the precision of the estimate by using all available information concerning performance and, at the same time, allows for handling missing data with more powerful modeling of the analysis. 

The LMMs were considered the two treatments: HILT + Hy-A as the reference group, the three times of the study, with baseline as the reference, and lastly, the interaction between time and treatment. Sensitivity analyses were also conducted, excluding from analysis those patients that did not reach a subjective and clinical improvement, applying the same previously described models. 

Data were analyzed with SAS software (Rel. 9.4, Cary, NC, USA), and the *p*-value for differences was considered statistically significant for a value less than or equal to 0.05.

## 3. Results

In this study, the medical records of 80 patients were included (40 for each group). No statistically significant between-group differences were found for main clinical characteristics, for peak and mean strength, and for the PRTEE-total score and PRTEE-subscales (Table 1).

Hand-grip peak-strength increased during the study in both groups, and a multiplicative effect was demonstrated for the interaction between treatment and time (*p* < 0.001) (Figure 1). 

In the HILT + Hy-A group, the peak strength significantly increased compared to the TE group in all follow-ups (baseline-T1: 11.21 ± 1.89; *p* < 0.001; baseline-T2: 12.06 ± 2.94; *p* < 0.001; baseline-T3: 7.57 ± 3.26; *p* = 0.02). Nearly one-quarter (ρ = 24.5, Appendix A, Model A) of the total variation in peak strength was attributable to differences between patients. Moreover, from the unconditional means model, 84% of unexplained residual in the variation of peak strength (Model A: 210.81 ± 18.49 Nw) was associated with linear time (Model B) and 87% with the interaction between time and treatment (Model C). Model goodness increased (AIC decreased between models) with model complexity. Mean strength increased during the study period in both groups, and again a multiplicative effect was demonstrated for the interaction between treatment and time (*p* < 0.001) (Figure 2). 

In the HILT + Hy-A group, the mean strength significantly increased compared to the TE group, only between baseline and T1 (17.24 ± 2.20; *p* < 0.001) and between baseline and T2 (14.49 ± 2.81; *p* < 0.001). Almost 15% (ρ = 14.5, Appendix A, Model A) of the total variation in peak strength was attributable to differences between patients. Again, Model B and Model C explained 84 and 89% of the unexplained residuals in the variation of peak strength (335.10 ± 29.21), respectively. Model goodness increased (AIC decreased between models) with model complexity.

Figure 3A–D report the PRTEE total score, the pain subscale score, the specific disability score, and the usual activity score. PRTEE total score and all the subscales decreased during the time points of the study. In the comparison between treatments according to different follow-ups, statistically significant differences were reported only for the comparisons between baseline and T1 and baseline and T2 for the PRTEE-score and all subscales. For the PRTEE score, almost 27% (ρ = 27.5, Appendix A, Model A) of the total variation was attributable to differences between patients. Again, Model B and Model C explained 72 and 76 percent of the unexplained residual in the variation of peak strength (83.93 ± 8.77), respectively. Model goodness increased (AIC decreased between models) with model complexity. After 6 months, 11 patients (13.8%), four in the HILT + HyA group and seven in the TE group (*p*-value = 0.33), did not improve.

## 4. Discussion

To our knowledge, no clinical studies investigated the effectiveness of the combined use of Hy-A and HILT on LET. Our findings suggest that combined peritendinous injections of Hy-A and HILT might be effective for improving pain, muscle strength, and disability in this population at short- and medium-term (1 to 3 month-follow-up) compared to TE.

Tendinopathy can occur because of different insults, such as certain drug treatments (e.g., fluoroquinolones), metabolic disorders (e.g., diabetes mellitus and hypercholesterolemia), and biomechanical factors [24]. In particular, overload and detraining are catabolic stimuli for tendon tissue resulting in increased synthesis of collagenase, proteinase, and pro-inflammatory cytokines [25]. To date, different studies support the use of eccentric exercise to improve pain and muscle strength in patients with LET [26]. Compared with the concentric exercise, the eccentric exercise showed a significant reduction in self-reported pain [26]. Eccentric contraction would appear to stimulate tendon cells, resulting in increased collagen cross-linking [27] and decreased neuro-vascular ingrowth that seem to modulate pain [28]. However, the relationship between exercise type and pain remains unclear in LET, and often, it is debated whether eccentric exercises should be performed with pain [29]. Other therapeutic strategies have been proposed in recent years for the treatment of pain in patients with LET, including injections of different agents, such as platelet-rich plasma (PRP), adipose-derived mesenchymal stromal cells, botulinum toxin, and Hy-A [30]. Hy-A injection seems to inhibit the pro-inflammatory response by local fibroblast [31], reduce pain, improve function, and reduce tendon rubbing in pre-insertion areas during major tendinopathies and post-surgical tendon repair [14]. Among adjunctive interventions for the management of patients with LET, physical modalities are commonly used in clinical practice [24]. In particular, LLLT seems to be useful in different musculoskeletal disorders by reducing edema and inflammation, controlling pain, and promoting tissue healing [32]. However, controversial evidence is available about the benefits of the use of LLLT in patients with LET, where only short-term pain relief was reported [33]. On the other hand, evidence about HILT is scant, with some observational studies suggesting the effectiveness of this intervention in patients with LET on pain control, functional recovery, and quality of life [34]. HILT might have analgesic and regenerative effects attributable to its ability to slow pain transmission, increase the production of morphine-mimetic substances [34], stimulate collagen production, and increase vascular permeability and blood flow within tendons by photochemical and photothermic stimulation [35]. These effects might act synergistically with the analgesic and regenerative effects of Hy-A. For what concerns safety, the combined approach proposed in our study was well tolerated, as demonstrated by the absence of AEs. Moreover, our data suggest that this approach promoted rapid clinical and functional improvement. Lastly, we cannot demonstrate a statistically significant reduction in the therapeutical failure between the two treatments, but this is not the main objective of the study, and probably the study is underpowered against this outcome.

The strengths of our study are the adequate sample size, the long-term follow-up, and no serious adverse effect occurrence in both study groups. Moreover, the character of a real-life study strengthens the results on the efficacy and safety of the combined treatment of Hy-A plus HILT in the treatment of LET. However, our study has some limitations. First, the retrospective design hampered the allocation of patients in the two groups because a randomization procedure was lacking. Moreover, a healthy worker effect could be introduced as further selection bias. In addition, our cohort might not be representative of the general population suffering from LET, such as elderly, overweight-obese people, and workers involved in repetitive and strenuous activities. The choice of a control group treated only with HILT, or alternatively, only with Hy-A, could be more informative to better define the role of each intervention in the control of LET symptomatology.

## 5. Conclusions

Our findings support the hypothesis that a multimodal approach might provide additional benefits without safety concerns in patients with LET. In particular, our study showed the effectiveness of the combined intervention of HILT and Hy-A injections over TE in subjects affected by LET in the short–medium term. Despite these encouraging findings, randomized controlled trials are required to prove the efficacy of the proposed approach compared to placebo or the efficacy of HILT + Hya + TE over the TE effect.

## Figures and Tables

**Figure 1 jcm-11-05492-f001:**
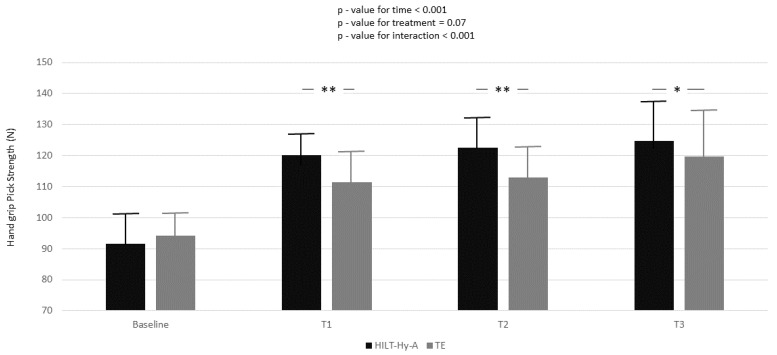
Handgrip peak strength variation according to time and treatment; black box represents high-intensity laser therapy plus hyaluronic acid (HILT + Hy-A) group, whereas grey box represents therapeutic exercise (TE). Results for the linear mixed model analysis (*p*-values for time, treatment, and interaction of time for treatment); Bonferroni adjustment for multiple comparisons were also applied between follow-ups of study compared to baseline; *: *p* < 0.05; **: *p* < 0.001.

**Figure 2 jcm-11-05492-f002:**
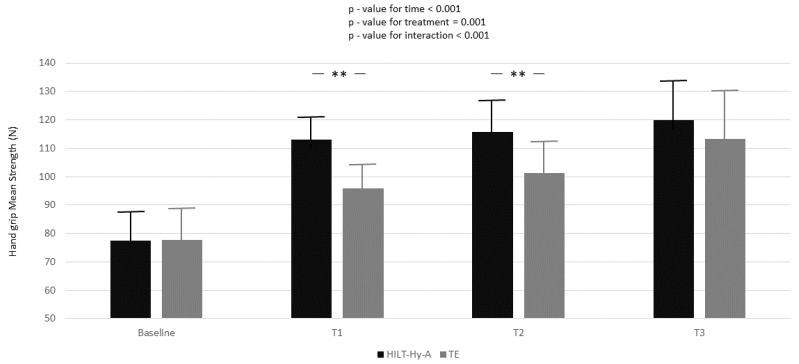
Handgrip mean strength variation according to time and treatment; black box represents high-intensity laser therapy plus hyaluronic acid (HILT+Hy-A) group, whereas grey box represents therapeutic exercise (TE). Results for the linear mixed model analysis (*p*-values for time, treatment, and interaction for time for treatment). Bonferroni adjustment for multiple comparisons applied between follow-ups of the study compared to baseline; **: *p* < 0.001.

**Figure 3 jcm-11-05492-f003:**
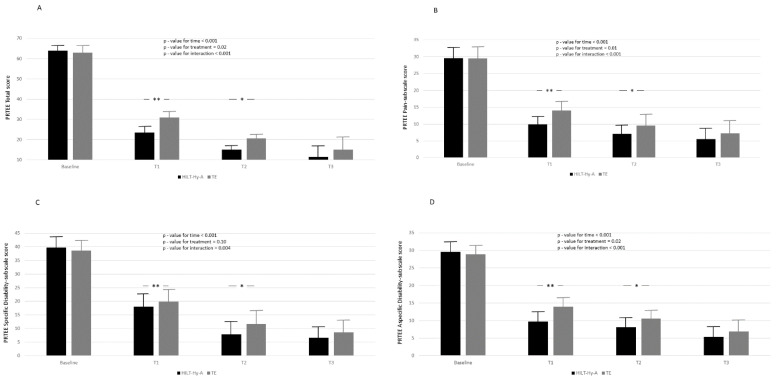
Report the PRTEE total score (**A**), the pain subscale score (**B**), the specific disabil-ity score (**C**), and the usual activity score (**D**). according to time and treatment; black box represents high-intensity laser therapy plus hyaluronic acid (HILT + Hy-A) group, whereas grey box represents therapeutic exercise (TE). Results for linear mixed model analysis (p-value for time, treatment, and interaction of time for reatment). Bonferroni adjustment for multiple comparisons applied between follow-ups of the study compared to baseline; *: *p* < 0.05, **: *p* < 0.001.

**Table 1 jcm-11-05492-t001:** Description of the population characteristics according to type of treatment at enrollment. High-intensity laser therapy plus hyaluronic acid (Hilt + Hy-A), therapeutic exercise (TE); Patient-Rated Tennis Elbow Evaluation (PRTEE).

	HILT + Hy-A	TE	*p*-Value
	40	40	
Age (year)	47.3 ± 9.5	50.3 ± 8.1	0.13
Sex female *n* (%)	25 (62.5)	21 (52.5)	0.37
Side pain (right) *n* (%)	19 (47.5)	22 (55.0)	0.50
Weight (kg)	74.1 ± 6.3	74.0 ± 5.5	0.97
Height (m)	1.68 ± 0.1	1.69 ± 0.1	0.75
BMI (kg/m^2^)	26.2 ± 2.3	26.1 ± 1.9	0.72
White collar *n* (%)	14 (35.0)	16 (40.0)	0.64
Education			0.78
Junior high school *n* (%)	16 (40.0)	13 (32.5)	
High school *n* (%)	15 (37.5)	17 (42.5)	
University degree *n* (%)	9 (22.5)	10 (25.0)	
Smoke habits (actually) *n* (%)	10 (25.0)	14 (35.0)	0.33
Hypertension *n* (%)	10 (25.0)	9 (22.5)	0.79
Cholesterol mmol/L	184.6 ± 29.7	184.0 ± 28.3	0.94
Hand grip Peak strength (Nw)	91.6 ± 12.2	94.1 ± 7.1	0.28
Hand grip Mean strength (Nw)	77.5 ± 10.0	77.7 ± 7.0	0.96
PRTEE total score	64.2 ± 6.2	63.3 ± 4.7	0.43
PRTEE Pain score	29.6 ± 3.4	29.5 ± 3.2	0.87
PRTEE Specific activities	39.7 ± 4.9	38.6 ± 3.4	0.28
PRTEE Usual activities	29.6 ± 2.8	28.9 ± 2.5	0.27

BMI: Body Mass Index.

## Data Availability

The datasets used and/or analyzed during the current study are available from the corresponding author upon reasonable request.

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
