# Peer review of "Effectiveness of High-Intensity Laser Therapy Plus Ultrasound-Guided Peritendinous Hyaluronic Acid Compared to Therapeutic Exercise for Patients with Lateral Elbow Tendinopathy"

_jcm, 2022, doi:10.3390/jcm11195492_

Round 1

Reviewer 1 Report

This retrospective study investigates the effectiveness of high-intensity laser therapy plus ultrasound- 2 guided peritendinous hyaluronic acid compared to therapeutic 3 exercise for patients with lateral elbow tendinopathy. I find the study interesting and well written.

I have few concerns.

Introduction

Lines 67-69: Need to substantiate/justify the preference of HILT over LLT. See Ahmad et al. (2022), which showed that HILT has higher efficacy in improving knee osteoarthritis-related outcomes than LLLT; considering knee osteoarthritis and lateral elbow tendinopathy both are musculoskeletal-related disorders, it is possible to anticipate similar effects/outcomes.

Reference

Ahmad et al. (2002). Effects of low-level and high-intensity laser therapy as adjunctive to rehabilitation exercise on pain, stiffness and function in knee osteoarthritis: a systematic review and meta-analysis. Physiotherapy114, 85–95. https://doi.org/10.1016/j.physio.2021.03.011

Materials and Methods

Describe who performed the medical records screening, sample selection

and allocation, and secondary data extraction (STROBE checklist item no

6).

Lines 123- 158: From my understanding, this is a retrospective longitudinal

study utilising secondary data from medical records; (i) is it possible that

participants in the HILT+HyA group also received some therapeutic exercise

as part of theirrehabilitation management (considering therapeutic exercise

as commonly prescribed care for LET)? Similarly, samples in the TE groups

also perhaps received physical/electro modalities such as hot/cold therapy,

electrical stimulation or therapeutic ultrasound as part of their treatment.

Please provide information.

Lines 142-158: Need a more detailed explanation of the therapeutic

exercise received by the participants grouped under TE; is it supervised,

home-based exercise or a combination? Are the prescribed exercises based

on recommended treatment guidelines for patients with lateral elbow

tendinopathy?

Results

Table 1

Change Kg to kg for Weight and BMI categories.

CHange Mmol to mmol for Cholesterol

Figures 1, 2 and 3 need to increase resolution and size. 

Conclusions

Lines 316-318: Suggest factor in therapeutic exercise for future study recommendations. Instead of “…randomized controlled trials are required to prove the efficacy of the proposed approach compared to placebo or other interventions”, may consider “HILT+HyA+Exercise compared to placebo (HILT+HyA)+Exercise”.

Author Response

Reviewer #1

This retrospective study investigates the effectiveness of high-intensity laser therapy plus ultrasound- 2 guided peritendinous hyaluronic acid compared to therapeutic 3 exercise for patients with lateral elbow tendinopathy. I find the study interesting and well written.

I have few concerns.

Introduction

  • Lines 67-69: Need to substantiate/justify the preference of HILT over LLT. See Ahmad et al. (2022), which showed that HILT has higher efficacy in improving knee osteoarthritis-related outcomes than LLLT; considering knee osteoarthritis and lateral elbow tendinopathy both are musculoskeletal-related disorders, it is possible to anticipate similar effects/outcomes. Ahmad et al. (2002). Effects of low-level and high-intensity laser therapy as adjunctive to rehabilitation exercise on pain, stiffness and function in knee osteoarthritis: a systematic review and meta-analysis. Physiotherapy, 114, 85–95. https://doi.org/10.1016/j.physio.2021.03.011

We agree with this reviewer, and accordingly we justify the preference, synthetizing only in one sentence with the reference suggested.

  • Materials and Methods: Describe who performed the medical records screening, sample selection and allocation, and secondary data extraction (STROBE checklist item no 6).

We thank the Reviewer, to point out this omission, and accordingly now the text look like: “at the Chiparo Physical Medicine and Rehabilitation Clinic in Lecce, Italy were consulted, and selected by a specialized nurse, blinded for the main outcomes of the study.”

  • Lines 123- 158: From my understanding, this is a retrospective longitudinal study utilizing secondary data from medical records; (i) is it possible that participants in the HILT+HyA group also received some therapeutic exercise as part of their rehabilitation management (considering therapeutic exercise as commonly prescribed care for LET)? Similarly, samples in the TE groups also perhaps received physical/electro modalities such as hot/cold therapy, electrical stimulation or therapeutic ultrasound as part of their treatment. Please provide information.

We thank this reviewer for his/her suggestion, this is an interesting point. As matter of fact, we would like to assess the role of HILT+HyA compared to TE, if we consider also different therapeutical approaches, we could not disentangle the “pure” role of the combined therapy compared to TE. Therefore, among the patients enrolled in our database were selected only those subjects who did not receive other physical therapy, and did not receive any other therapeutical approach (line144-146). We also agree that TE is the first line approach to LET, but in the real word, in several situation patients did not agree to consider TE as an integrative approach. This happens for example, in those subjects that tried TE in the past as the first line intervention, or they want something less time consuming. From those observations we started our analysis.

  • Lines 142-158: Need a more detailed explanation of the therapeutic exercise received by the participants grouped under TE; is it supervised, home-based exercise or a combination? Are the prescribed exercises based on recommended treatment guidelines for patients with lateral elbow tendinopathy?

We thank the Reviewer, for pointing out this “omissions”: 1) due to the semiautomated software that handle bibliography, was not reported the adequate reference 2) moreover, we clearly stated that patients were in clinical supervised

  • Results: Table 1, Change Kg to kg for Weight and BMI categories. Change Mmol to mmol for Cholesterol. Figures 1, 2 and 3 need to increase resolution and size.

We thank the reviewer, and accordingly we emended the table. Figures, we increased the resolution and size.

  • Conclusions: Lines 316-318: Suggest factor in therapeutic exercise for future study recommendations. Instead of “…randomized controlled trials are required to prove the efficacy of the proposed approach compared to placebo or other interventions”, may consider “HILT+HyA+Exercise compared to placebo (HILT+HyA)+Exercise”.

Accordingly with this referee suggestion we modify the last sentence of the conclusion that now looks like:

“Despite these encouraging findings, randomized controlled trials are required to prove the efficacy of the proposed approach compared to placebo or the efficacy of HILT+Hya+TE over the pure TE effect.

Reviewer 2 Report

Add the other modalities of management of tennis elbow. 

Is hyaluronic acid + high intensity laser superior to platelet rich plasma, stem cells injections?

Any radiological evaluation done pre and post procedure?

Follow up duration is short. Atleast 6 months follow up is needed. 

Minor language polishing needed. 

Add a few clinical images 

Author Response

Reviewer #2

  • Add the other modalities of management of tennis elbow.

We thank the Reviewer and accordingly we add conservative and surgery approaches.

2)        Is hyaluronic acid + high intensity laser superior to platelet rich plasma, stem cells injections?

       We could not answer to this question, at least in this paper. Our protocol did not consider PRP or stem cells injection. We thank the reviewer for the suggestion, and we will consider this problem in next research.

3)        Any radiological evaluation done pre and post procedure?

       As we stated in the Method section, all patients were assessed with: “Ultrasound (US) evaluation showing thickening and heterogeneous eco structure of the common extensor tendon, as well as increased blood flow under Doppler”.  (lines 102-104)

4)        Follow up duration is short. At least 6 months follow up is needed.

We agree with this Reviewer, and as we have stated in the first version of the paper, we followed patients for at least six months Outcomes section:

“The same functional assessments were performed at different follow-ups: T1 (30 days), T2 (90 days), and T3 (180 days) after the end of treatment.”

5)        Minor language polishing needed.

We thank the reviewer and accordingly we revised the paper.

6)        Add a few clinical images

Accordingly with this referee suggestion we include in the paper two photo in supplementary material: Supplementary Figure 1 US image of the LET with an insertional calcification; Supplementary Figure 2 HyA-US-guided injection.